# Fast Sparse Group Lasso

**Yasutoshi Ida**[1,3]    **Yasuhiro Fujiwara**[2]    **Hisashi Kashima**[3,4]
[1]NTT Software Innovation Center    [2]NTT Communication Science Laboratories
[3]Kyoto University    [4]RIKEN AIP
yasutoshi.ida@ieee.org
yasuhiro.fujiwara.kh@hco.ntt.co.jp
kashima@i.kyoto-u.ac.jp

## Abstract

Sparse Group Lasso is a method of linear regression analysis that finds sparse parameters in terms of both feature groups and individual features. Block Coordinate Descent is a standard approach to obtain the parameters of Sparse Group Lasso, and iteratively updates the parameters for each parameter group. However, as an update of only one parameter group depends on all the parameter groups or data points, the computation cost is high when the number of the parameters or data points is large. This paper proposes a fast Block Coordinate Descent for Sparse Group Lasso. It efficiently skips the updates of the groups whose parameters must be zeros by using the parameters in one group. In addition, it preferentially updates parameters in a candidate group set, which contains groups whose parameters must not be zeros. Theoretically, our approach guarantees the same results as the original Block Coordinate Descent. Experiments show that our algorithm enhances the efficiency of the original algorithm without any loss of accuracy.

## 1 Introduction

Sparse Group Lasso (SGL) [3, 17] is a popular feature-selection method based on the linear regression model for data that have group structures. For the analysis of such data, it is important to identify not only individual features but also groups of features that have some relationships with the response. SGL finds such groups and features by obtaining sparse parameters corresponding to the features in the linear regression model. In particular, SGL effectively achieves parameter sparsity by utilizing two types of regularizations: feature- and group-level regularization. Owing to its effectiveness, SGL is used in the analysis of the various data, e.g., gene expression data [11, 16] and climate data [13].

In order to obtain the sparse parameters in SGL, Block Coordinate Descent (BCD) is used as a standard approach [3, 17]. BCD iteratively updates the parameters for each group until convergence. In particular, it first checks whether the parameters in a group are zeros by using all the parameters or data points. This process induces group-level sparsity. If the parameters in the group are determined as nonzeros, BCD updates the parameters in the group. It applies the aforementioned steps to the parameters of each group until the parameters of all groups converge.

Although SGL is practical for analyzing group structured data, BCD suffers from high computation costs. The main bottleneck is the computation to check whether the parameters of a group are zeros, because the computation uses all the parameters or data points. The screening technique is the main existing approach for reducing the computation cost of BCD by reducing the data size [18, 12–14]. This technique eliminates features and groups whose parameters are zeros, before entering the iterations of BCD. However, the screening techniques cannot be expected to reduce the data size when the initial parameters are far from optimal [9]. The screening techniques often face such problems in practice, and the efficiency of BCD would not be increased in such cases. Therefore, speeding up BCD is still an important topic of study for handling large data sizes.

This paper proposes a fast BCD for SGL. Our main idea is to identify the groups whose parameters *must be zeros* by only using the parameters in the group, whereas the standard method uses all the parameters or data points. As the number of parameters in one group is much smaller than the total number of parameters or data points, our method efficiently skips the computation of groups whose parameters must be zeros. Another idea is to extract a *candidate group set*, which contains groups whose parameters *must not be zeros*. As the parameters in the set are likely to largely contribute to the prediction [5, 4], we can expect BCD to effectively optimize the objective function by preferentially updating the parameters in the set. The attractive point of our method is that it does not need any additional hyperparameters, which incur additional tuning costs. In addition, it provably guarantees convergence to the same value of the objective function as the original BCD. Experiments demonstrate that our method enhances the efficiency of BCD while achieving the same prediction error. Although we consider the case of non-overlapping groups in the paper, our method is relatively easy to be extended for overlapping groups by using overlap norm [8].

## 2  Preliminary

### 2.1  Sparse Group Lasso

This section defines the SGL as a method of linear regression analysis that finds small sets of groups in addition to features that achieve high prediction accuracy for the response. Let $n$ be the number of data points, where each data point is represented as a $p$-dimensional feature vector, $y \in \mathbb{R}^n$ be a continuous response, and $G$ be the total number of feature groups. A matrix of features $X \in \mathbb{R}^{n \times p}$ is represented as $X = [X^{(1)}, X^{(2)}, ..., X^{(G)}]$, where $X^{(g)} \in \mathbb{R}^{n \times p_g}$ is the block matrix of $X$ corresponding to the $g$-th feature group with the number of features $p_g$. Similarly, parameter vector $\beta \in \mathbb{R}^p$ is represented as $\beta = [\beta^{(1)\mathrm{T}}, \beta^{(2)\mathrm{T}}, ..., \beta^{(G)\mathrm{T}}]^{\mathrm{T}}$, where $\beta^{(g)} \in \mathbb{R}^{p_g}$ is the parameter (coefficient) vector of group $g$. Therefore, the linear regression model in SGL is represented as $y = X\beta = X^{(1)}\beta^{(1)} + \cdots + X^{(G)}\beta^{(G)}$. Solution $\hat{\beta}$ is obtained by solving the following problem:

$$\hat{\beta} = \underset{\beta \in \mathbb{R}^p}{\arg\min} \frac{1}{2n}\|y - \sum_{g=1}^{G} X^{(g)}\beta^{(g)}\|_2^2 + (1-\alpha)\lambda \sum_{g=1}^{G} \sqrt{p_g}\|\beta^{(g)}\|_2 + \alpha\lambda\|\beta\|_1, \qquad (1)$$

where $\alpha \in [0, 1]$ and $\lambda \geq 0$ are regularization constants; $\alpha$ decides the balances of the convex combination of $l_1$ and $l_2$ norm penalties and $\lambda$ controls the degree of sparsity of the solution.

### 2.2  Block Coordinate Descent

BCD is a standard approach used to obtain solution $\hat{\beta}$ of SGL [17]. It consists of a group-level outer loop and an element-level loop. The group-level outer loop checks whether parameter vector $\beta^{(g)}$ for each feature group is a zero vector. If $\beta^{(g)}$ turns to be a nonzero vector, the element-level loop updates each parameter in $\beta^{(g)}$. The process terminates if the whole parameter vector converges.

In the element-level loop, BCD updates $\beta^{(g)}$ in group $g$ if the parameter vector of the group is not a zero vector. The updated parameter vector $\beta_{\mathrm{new}}^{(g)}$ is defined as follows:

$$\beta_{\mathrm{new}}^{(g)} = \left(1 - \frac{t(1-\alpha)\lambda}{\|S(Z^{(g)}, t\alpha\lambda)\|_2}\right)_+ S(Z^{(g)}, t\alpha\lambda). \qquad (2)$$

In Equation (2), $Z^{(g)} = \beta^{(g)} + \frac{t}{n}(X^{(g)\mathrm{T}}r_{(-g)} - X^{(g)\mathrm{T}}X^{(g)}\beta^{(g)})$, where $t \geq 0$ is the step size, and $r_{(-g)}$ is the partial residual and is defined as follows:

$$r_{(-g)} = y - \sum_{l \neq g}^{G} X^{(l)}\beta^{(l)}. \qquad (3)$$

$S(\cdot)$ is the coordinate-wise soft-threshold operator; the $j$-th element is computed as $S(z, \gamma)[j] = \mathrm{sign}(z[j])(|z[j]| - \gamma)_+$. $\beta^{(g)}$ is iteratively updated using Equation (2) until convergence. If the parameter vector of the group is determined as a zero vector, Equation (2) is skipped. The computation cost of Equation (2) is $\mathcal{O}(p_g^2)$ time because $X^{(g)\mathrm{T}}X^{(g)}$ in $Z^{(g)}$ are precomputed before entering the main loop. In addition, $X^{(g)\mathrm{T}}r_{(-g)}$ has already been computed in the group-level outer loop, as described next.

In the group-level outer loop, $\beta^{(g)}$ is the zero vector if the following condition holds:

$$\|S(X^{(g)\mathrm{T}}r_{(-g)}, \alpha\lambda)\|_2 \leq \sqrt{p_g}(1-\alpha)\lambda. \qquad (4)$$

In other words, if Equation (4) holds, the parameter vector of group $g$ is a zero vector; Equation (2) is then skipped, and the parameter vector is not updated. $X^{(g)\mathrm{T}}r_{(-g)}$ in Equation (4) is computed using the following equation that consists of only matrix operations:

$$X^{(g)\mathrm{T}}r_{(-g)} = X^{(g)\mathrm{T}}y - X^{(g)\mathrm{T}}X\beta + X^{(g)\mathrm{T}}X^{(g)}\beta^{(g)}. \tag{5}$$

In this equation, $X^{(g)\mathrm{T}}y$, $X^{(g)\mathrm{T}}X$, and $X^{(g)\mathrm{T}}X^{(g)}$ are precomputed before entering the loops. The costs of the precomputations have relatively low impacts on the total computational cost because precomputations are performed only once in the total computation, and are easily parallelized. On the other hand, the computation cost of Equation (5) is still high because it requires $\mathcal{O}(pp_g + p_g^2)$ time, and it is repeatedly performed until convergence. As a result, we need $\mathcal{O}(pp_g + p_g^2)$ time for Equation (4) at every iteration. We can modify Equation (5) to have $\mathcal{O}(np_g)$ time by maintaining the partial residuals of Equation (3) as described in [7]. However, in either case, the computation cost of Equation (4) depends on $p$ or $n$. Therefore, Equation (4) incurs a large computation cost for high-dimensional features or a large number of data points.

## 3 Proposed Approach

In this section, we introduce our algorithm, which efficiently obtains the solution of SGL. First, we explain the ideas underlying our algorithm. Next, we introduce several lemmas that are necessary to derive our algorithm. We then describe the algorithm.[1]

### 3.1 Idea

In SGL, obtaining the solution through BCD incurs a high computation cost. This is because (i) Equation (4) requires $\mathcal{O}(pp_g + p_g^2)$ or $\mathcal{O}(np_g)$ time, which incurs a large computation cost for large feature vectors or a large number of data points, and (ii) BCD always checks all of the feature groups using Equation (4) at every iteration even when most of the groups have zero vectors.

Our main idea is to identify groups whose parameter vectors *must be zero vectors* by approximating Equation (4), which checks whether the parameter vector of each group is a zero. In particular, we compute the upper bound of $\|S(X^{(g)\mathrm{T}}r_{(-g)}, \alpha\lambda)\|_2$ instead of computing the exact value. If the upper bound is lower than $\sqrt{p_g}(1-\alpha)\lambda$, the parameter vector of the group must be a zero vector. As a result, we can safely skip the computation of the group. As our upper bound requires only $\mathcal{O}(p_g)$ time instead of the $\mathcal{O}(pp_g + p_g^2)$ or $\mathcal{O}(np_g)$ time for the original Equation (4), we can effectively reduce the computation cost.

Another idea is to extract a *candidate group set*, which contains groups whose parameter vectors *must not be zero vectors*. As the parameters in the set are likely to largely contribute to the prediction [5, 4], we can expect BCD to effectively optimize the objective function by preferentially updating the parameters in the set. In addition, our method only requires $\mathcal{O}(G)$ time to construct the set, and thus the computation cost is relatively low.

### 3.2 Upper Bound for Skipping Computations

We introduce the upper bound of $\|S(X^{(g)\mathrm{T}}r_{(-g)}, \alpha\lambda)\|_2$ in Equation (4). To derive a tight upper bound, we introduce reference parameter vectors and partial residuals of Equation (3) that are computed before entering the group-level outer loop. To be specific, we can obtain a tight bound by explicitly utilizing the term representing the difference between the reference and current parameter vectors. As many parameter vectors rapidly converge during the iterations, the difference between the reference and current parameter vectors rapidly decreases. We define the upper bound as follows:

**Definition 1 (Upper bound)** *Let $U^{(g)}$ be an upper bound of $\|S(X^{(g)\mathrm{T}}r_{(-g)}, \alpha\lambda)\|_2$ in Equation (4), and $\tilde{r}_{(-g)}$ be a partial residual of Equation (3) before entering the group-level outer loop. Then, $U^{(g)}$ is defined as follows:*

$$U^{(g)} = \|X^{(g)\mathrm{T}}\tilde{r}_{(-g)}\|_2 + \Lambda(g,g) + \sum_{l=1}^{G}\Lambda(g,l), \tag{6}$$

where $\Lambda(g,l) = \|\hat{K}^{(g)}[l]\|_2\|\beta^{(l)} - \tilde{\beta}^{(l)}\|_2$. *The $i$-th element of $\hat{K}^{(g)}[l] \in \mathbb{R}^{p_g}$ is given as $\|K^{(g,l)}[i,:]\|_2$, that is, the $l_2$ norm of the $i$-th row vector in block matrix $K^{(g,l)} \in \mathbb{R}^{p_g \times p_l}$ of $K := X^{\mathrm{T}}X \in \mathbb{R}^{p \times p}$. $\tilde{\beta}^{(g)}$ is a parameter vector before entering the group-level outer loop.*

Note that we can precompute $\|X^{(g)\mathrm{T}}\tilde{r}_{(-g)}\|_2$ and $\|\hat{K}^{(g)}[\cdot]\|_2$ before entering the group-level outer loop and the main loop, respectively. The following lemma shows the property of the upper bound corresponding to groups with parameters that must be zeros:

**Lemma 1 (Groups with zero vectors)** *If $U^{(g)}$ satisfies $U^{(g)} \le \sqrt{p_g}(1-\alpha)\lambda$, parameter $\beta^{(g)}$ for group $g$ is a zero vector.*

Lemma 1 indicates that we can identify groups whose parameters must be zeros by using upper bound $U^{(g)}$ instead of $\|S(X^{(g)\mathrm{T}}r_{(-g)}, \alpha\lambda)\|_2$. The error bound of $U^{(g)}$ for $\|S(X^{(g)\mathrm{T}}r_{(-g)}, \alpha\lambda)\|_2$ is described in a later section.

### 3.3  Online Update Scheme of Upper Bound

Although we can identify groups whose parameters must be zeros by using upper bound $U^{(g)}$, $\mathcal{O}(p + p_g)$ time is still required to compute Equation (6) of the upper bound even if we precompute $\|X^{(g)\mathrm{T}}\tilde{r}_{(-g)}\|_2$ and $\|\hat{K}^{(g)}[\cdot]\|_2$. As the standard approach requires $\mathcal{O}(pp_g + p_g^2)$ or $\mathcal{O}(np_g)$ time, the efficiency of our approach would be moderate. This is the motivation behind our use of the online update scheme for the upper bound. In particular, when a parameter vector of a group is updated, we use the following definition for the upper-bound computation:

**Definition 2 (Online update scheme of upper bound)** *If $\beta^{(g)}$ is updated to $\beta^{(g)'}$, we update upper bound $U^{(g)}$ of Equation (6) as follows:*

$$U^{(g)'} = U^{(g)} - 2\Lambda(g,g) + 2\|\hat{K}^{(g)}[g]\|_2\|\beta^{(g)'} - \tilde{\beta}^{(g)}\|_2. \tag{7}$$

Equation (7) clearly holds because we subtract old values of $2\Lambda(g,g)$ from Equation (6), and add updated values of $2\|\hat{K}^{(g)}[g]\|_2\|\beta^{(g)'} - \tilde{\beta}^{(g)}\|_2$ to the equation. In terms of the computation cost, we have the following lemma:

**Lemma 2 (Computation cost for online update scheme of upper bound)** *The computation of Equation (7) requires $\mathcal{O}(p_g)$ time given precomputed $\|X^{(g)\mathrm{T}}\tilde{r}_{(-g)}\|_2$ and $\|\hat{K}^{(g)}[\cdot]\|_2$ when the parameter vector of group $g$ is updated.*

The above lemma shows that we can update the upper bound in $\mathcal{O}(p_g)$ time. The computation cost is significantly low compared with the computations of Equations (4) and (6), which require $\mathcal{O}(pp_g + p_g^2)$ (or $\mathcal{O}(np_g)$) and $\mathcal{O}(p + p_g)$ times, respectively. Therefore, we can efficiently identify groups whose parameters must be zeros on the basis of Lemma 1 and Definition 2.

### 3.4  Candidate Group Set for Selective Updates

In this section, we introduce a method to extract the *candidate group set*, which contains the groups whose parameters must *not* be zeros. We expect BCD to effectively update the parameter vectors by preferentially updating the parameter vectors on the candidate group set. To extract the candidate group set, we utilize a criterion, which approximates $\|S(X^{(g)\mathrm{T}}r_{(-g)}, \alpha\lambda)\|_2$ in Equation (4). If the criterion, defined as follows, is above a threshold, the group is included in the set.

**Definition 3 (Criterion to extract candidate group set)** *Let $C^{(g)}$ be a criterion, which is used to check whether the group is included in the candidate group set. Then, $C^{(g)}$ is defined as follows:*

$$C^{(g)} = \|X^{(g)\mathrm{T}}\tilde{r}_{(-g)}\|_2 - \alpha\lambda\sqrt{p_g/2}, \tag{8}$$

*where $\tilde{r}_{(-g)}$ is a partial residual of Equation (3) before entering the group-level outer loop.*

The error bounds of $C^{(g)}$ and $U^{(g)}$ for $\|S(X^{(g)\mathrm{T}}r_{(-g)}, \alpha\lambda)\|_2$ are shown as follows:

**Lemma 3 (Error bound)** *Let $\epsilon$ be an error bound of $C^{(g)}$ for $\|S(X^{(g)\mathrm{T}}r_{(-g)}, \alpha\lambda)\|_2$ such that $|C^{(g)} - \|S(X^{(g)\mathrm{T}}r_{(-g)}, \alpha\lambda)\|_2| \leq \epsilon$. We then have $\epsilon = \Lambda(g, g) + \sum_{l=1}^{G} \Lambda(g, l) + \alpha\lambda\sqrt{p_g/2}$. In addition, we have $|U^{(g)} - \|S(X^{(g)\mathrm{T}}r_{(-g)}, \alpha\lambda)\|_2| \leq 2\epsilon$.*

The above lemma suggests that $C^{(g)}$ approximates $\|S(X^{(g)\mathrm{T}}r_{(-g)}, \alpha\lambda)\|_2$ better than $U^{(g)}$ because the error bound of $C^{(g)}$ is half the size of that of $U^{(g)}$. We extract candidate group set $\mathbb{C}$ with respect to $C^{(g)}$ by using the following definition:

**Definition 4 (Candidate group set)** *Candidate group set $\mathbb{C}$ is defined as*

$$\mathbb{C} = \{g \in \{1, ..., G\} | C^{(g)} > \sqrt{p_g}(1-\alpha)\lambda\}. \tag{9}$$

The candidate group set has the following property:

**Lemma 4 (Groups containing nonzero vectors)** *Candidate group set $\mathbb{C}$ contains the groups whose parameters must be nonzeros.*

The above lemma suggests that the candidate group set comprises not only the groups whose parameters must be nonzeros but also groups whose parameters can be nonzeros. In terms of the computation cost, we have the following lemma to extract the candidate group set:

**Lemma 5 (Computation cost of candidate group set)** *Given precomputed $\|X^{(g)\mathrm{T}}\tilde{r}_{(-g)}\|_2$, we can extract candidate group set $\mathbb{C}$ at $\mathcal{O}(G)$ time.*

### 3.5 Algorithm

This section describes our algorithm, which utilizes the above-mentioned definitions and lemmas. Algorithm 1 gives a full description of our approach, which is based on BCD with the sequential rule [6]: a standard approach for SGL. The sequential rule is used to tune regularization constant $\lambda$ with respect to the sequence of $(\lambda_q)_{q=0}^{Q-1}$, where $\lambda_0 > \lambda_1 > ... > \lambda_{Q-1}$: it sequentially optimizes the parameter vector by using $(\lambda_q)_{q=0}^{Q-1}$, and reuses the solution of the previous $\lambda$ as the initial parameters for the current $\lambda$.

Our main idea is to skip groups whose parameters must be zeros during the optimization by utilizing Lemma 1. As upper bound $U^{(g)}$ in Lemma 1 can be computed with a low computation cost as described in Lemma 2, we can efficiently avoid the computation of Equation (4), which is the main bottleneck of standard BCD. In addition, we extract the candidate group set before we start to optimize the parameters for the current $\lambda$. The impact of the computation cost is relatively low on the total cost, as shown in Lemma 5. We expected BCD to raise the effectiveness by preferentially updating the parameters in the set based on Lemma 4.

In Algorithm 1, (lines 2–4), first precomputes $\|\hat{K}^{(g)}[l]\|_2$, which is used for computing the upper bounds. In the loop of the sequential rule, we construct the candidate group set (lines 6–10). Although we computed Equation (9) in the initial iteration, we reused the term $\|X^{(g)\mathrm{T}}\tilde{r}_{(-g)}\|_2$ of the previous iteration in the equation for the other iterations. Next, BCD is performed on the parameter vectors of the set (lines 11–19). Then, the algorithm enters the loop of another BCD with upper bounds (lines 20–36). The reference parameter vector is set (line 21), and $\|X^{(g)\mathrm{T}}\tilde{r}_{(-g)}\|_2$ is precomputed, which is also used for the computation of the upper bounds (lines 22 and 23). In the group-level outer loop, upper bound $U^{(g)}$ of group $g$ was computed using Equation (7) (line 25). Note that Equation (6) is used for the initial computation of the upper bound. If bound $U^{(g)}$ is lower than threshold $\sqrt{p_g}(1-\alpha)\lambda$, the parameters of the group were set to zeros by following Lemma 1 (lines 26 and 27). If the bound does not meet the condition, the same procedure as that of the original BCD is performed (lines 28–34). Next, $\|\beta^{(g)} - \tilde{\beta}^{(g)}\|_2$, which is used for the computation of the upper bound is updated (line 35).

In terms of the computation cost, our algorithm has the following property:

**Theorem 1 (Computation cost)** *Let $S$ and $S'$ be the rates of the un-skipped groups when Lemma 1 and Equation (4) are used, respectively. Suppose that all groups have the same size, $p_g$. If $t_m$ and*

**Algorithm 1** Fast Sparse Group Lasso

```
 1:  𝔸 = {1, ..., G}, β ← 0, β̃ ← 0;                                                    ▷ 𝔸 has all the group indices
 2:  for each g ∈ 𝔸 do                                                                ▷ The precomputation for the bounds
 3:  │  for each l ∈ 𝔸 do
 4:  │  │  compute ‖K̂^(g)[l]‖₂;
 5:  for q = 0 to Q − 1 do                                    ▷ The loop for the sequential rule of regularization constants (λ_q)_{q=0}^{Q-1}
 6:  │  ℂ = ∅;                                                                         ▷ Initialize candidate group set ℂ
 7:  │  for each g ∈ 𝔸 do                                                             ▷ The loop for extracting candidate group set
 8:  │  │  compute C^(g) by Equation (8);
 9:  │  │  if C^(g) > √p_g(1 − α)λ_q then                                              ▷ Add groups to ℂ by following Lemma 4
10:  │  │  │  add g to ℂ;
11:  │  repeat                                                                         ▷ The main loop for BCD on candidate group set ℂ
12:  │  │  for each g ∈ ℂ do                                                          ▷ Group-level outer loop
13:  │  │  │  if ‖S(X^(g)T r_{(−g)}, αλ_l)‖₂ ≤ √p_g(1 − α)λ_q then                     ▷ Check the condition of Equation (4)
14:  │  │  │  │  β^(g) ← 0;
15:  │  │  │  else
16:  │  │  │  │  repeat                                                                ▷ Element-level loop
17:  │  │  │  │  │  update β^(g) by Equation (2);
18:  │  │  │  │  until β^(g) converges
19:  │  until β converges
20:  │  repeat                                                                         ▷ The main loop for BCD with the upper bounds
21:  │  │  β̃ ← β;                                                                      ▷ Set the reference parameter vector
22:  │  │  for each g ∈ 𝔸 do                                                          ▷ The precomputation for the upper bounds
23:  │  │  │  compute ‖X^(g)T r̃_{(−g)}‖₂;
24:  │  │  for each g ∈ 𝔸 do                                                          ▷ Group-level outer loop
25:  │  │  │  compute U^(g) by Equation (7);
26:  │  │  │  if U^(g) ≤ √p_g(1 − α)λ_q then                       ▷ Skip the group whose parameters must be zeros by following Lemma 1
27:  │  │  │  │  β^(g) ← 0;
28:  │  │  │  else
29:  │  │  │  │  if ‖S(X^(g)T r_{(−g)}, αλ_l)‖₂ ≤ √p_g(1 − α)λ_q then                  ▷ Check the condition of Equation (4)
30:  │  │  │  │  │  β^(g) ← 0;
31:  │  │  │  │  else
32:  │  │  │  │  │  repeat                                                             ▷ Element-level loop
33:  │  │  │  │  │  │  update β^(g) by Equation (2);
34:  │  │  │  │  │  until β^(g) converges
35:  │  │  │  update ‖β^(g) − β̃^(g)‖₂;                                                ▷ For online update scheme for the upper bounds
36:  │  until β converges
```

$t_f$ are the numbers of iterations of BCD for the main loop and element-level loop, respectively, our approach requires $\mathcal{O}(G\{(Q + St_m)(pp_g + p_g^2) + S'p_g t_m(t_f p_g + 1) + Q\})$ or $\mathcal{O}(G\{(Q + St_m)np_g + S'p_g t_m(t_f p_g + 1) + Q\})$ time.

According to Theorem 1, when we have a large number of groups that are skipped on the basis of the upper bound, the rate of un-skipped groups $S$ in Theorem 1 is small. As a result, the total computation cost is effectively reduced.

In terms of the convergence, our algorithm has the following property:

**Theorem 2 (Convergence property)** *Suppose that the regularization constants in Algorithm 1 are the same as those of the original BCD, and the BCD converges. Then, the solution of Algorithm 1 has the same value of the objective function as that of the original BCD.*

Theorem 2 shows that our algorithm returns the same value of the objective function as the original approach. Therefore, our approach dose not decrease the accuracy compared to the original approach.

## 4  Related Work

To improve the efficiency of optimization with sparsity-inducing regularization, safe screening is generally used [6]; it eliminates zero parameters in the solution before the optimization is initiated. As the size of the feature vector can be reduced before optimizing the problem, the efficiency of the optimization is improved. The current state-of-the-art safe screening method for SGL is the GAP

Safe rule [18, 12–14], which is based on dual gap computation. The dual gap is computed as the difference between the primal and dual problems of SGL. They define a safe region that contains the solution based on the dual gap. By utilizing the safe region, this approach can identify groups and features that must be inactive, and eliminates them. If the safe region is small, this approach effectively eliminates groups and features. However, unless $\lambda$ is large or a good approximate solution is already known, the screening is often ineffective [9]. To overcome this problem, Ndiaye et al. [14] used the dynamic safe rule [1, 2] with the GAP Safe rule for SGL. This dynamic GAP Safe rule effectively eliminates groups and features by repeatedly using the GAP Safe rule during the iterations of BCD.

## 5 Experiments

We evaluated the processing time and prediction error of our approach by conducting experiments on six datasets from the LIBSVM[2] website (*abalone*, *cpusmall*, *boston*, *bodyfat*, *eunite2001*, and *pyrim*). The numbers of data points were 4177, 8192, 506, 252, 336, and 74, respectively. In order to obtain group structure, we used the polynomial features of these datasets [15]. In particular, we created second-order polynomial features by following the method used in [16]. The groups, which consisted of product over combinations of features up to the second degree, were created by using a polynomial kernel. As a result, the numbers of groups for each dataset were 36, 78, 91, 105, 136, and 378, respectively. The total numbers of features were 176, 408, 481, 560, 736, and 2133, respectively.

We compared our method with the original BCD, GAP Safe rule [13], and dynamic GAP Safe rule [14]. We tuned $\lambda$ for all approaches based on the sequential rule by following the methods in [18, 12–14]. The search space was a non-increasing sequence of $Q$ parameters $(\lambda_q)_{q=0}^{Q-1}$ defined as $\lambda_q = \lambda_{max} 10^{-\delta q/Q-1}$. We used $\delta = 4$ and $Q = 100$ [18, 12–14]. For a fair comparison, $\lambda_{max}$ was computed according to the dual norm by following the concept of GAP Safe rule [13]; Gap Safe rule safely eliminates groups and features under this setting. For dynamic GAP Safe rule, the interval of dual gap computations is set to 10 [14]. For another tuning parameter $\alpha$, we used the settings $\alpha \in [0.2, 0.4, 0.6, 0.8]$. We stopped the algorithm for each $\lambda_q$ when the relative tolerance $\|\beta - \beta_{new}\|_2 / \|\beta_{new}\|_2$ dropped below $10^{-5}$ for all approaches [9, 10]. All the experiments were conducted on a Linux 2.20 GHz Intel Xeon server with 264 GB of main memory.

### 5.1 Processing Time

We evaluated the processing times of the sequential rules for each $\alpha \in [0.2, 0.4, 0.6, 0.8]$. Figure 1 shows the processing time of each approach on the six datasets. Note that the processing times include precomputation times for a fair comparison. In the figure, the terms *origin*, *GAP*, *dynamic GAP*, and *ours* represent the standard BCD, GAP Safe rule [13], dynamic GAP Safe rule [14], and our approach, respectively. Our approach is faster than the previous approaches for all datasets and hyperparameters; it reduces the processing time by up to $97\%$ from the standard approach as shown in Figure 1 (f). Table 1 shows the number of computations for Equation (4), which is the main bottleneck of BCD. The result suggests the effectiveness of the upper bound and candidate group set, which effectively reduce the number of computations, and contribute to the reduction of the processing time, as shown in Figure 1. The GAP Safe rule and dynamic GAP Safe rule eliminate groups and features that must be inactive, and increase the efficiency of BCD. However, when they cannot eliminate a significant number of groups and features, they require a large computation cost for BCD. To be specific, large numbers of groups and features remain when $\lambda$ has a small value even if we use dynamic GAP Safe rule. This is because the safe region is large for small $\lambda$ [12, 13], and it contains many groups and features that may be active. Furthermore, if the screening cannot eliminate a significant number of groups and features, the processing time may increase owing to the computation of the dual gap, as shown for $\alpha = 0.4$ in Figure 1(a).

### 5.2 Accuracy

In this section, we evaluate the prediction error on test data to confirm the effectiveness of our algorithm. We split the data into training and test data for each dataset. That is, $50\%$ of a dataset was used as test data for evaluating the prediction error in terms of the squared loss for the response. The

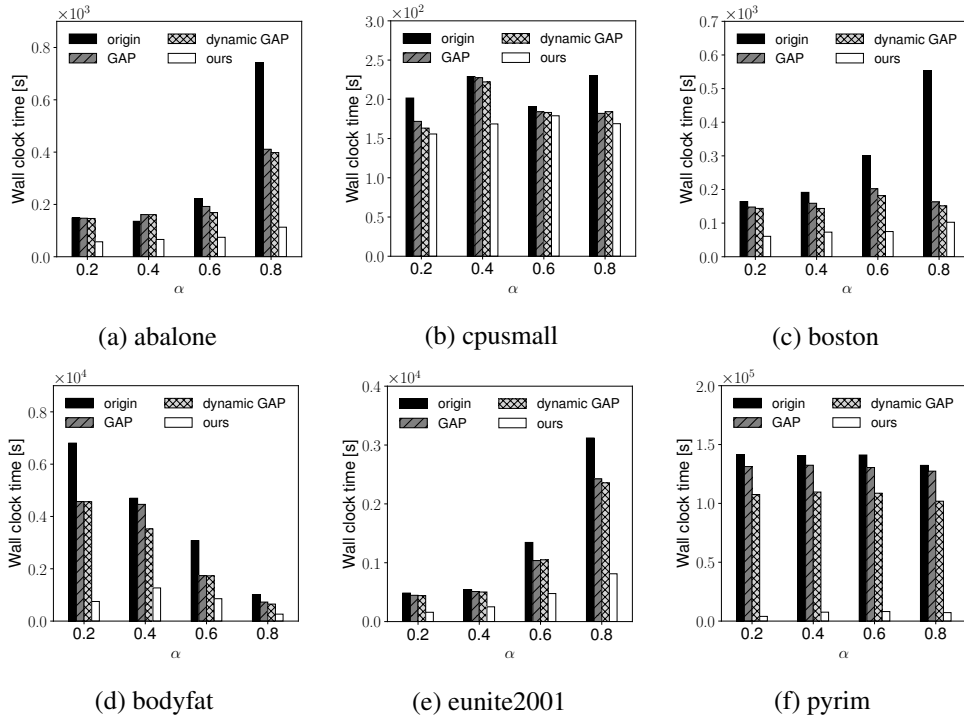

Figure 1: Processing times of sequential rules for each hyperparameter $\alpha$.

Table 1: Numbers of computations for Eq. (4)

| dataset | # of computations for Eq. (4) | |
| --- | --- | --- |
| | origin | ours |
| abalone | $1.141 \times 10^5$ | $\mathbf{3.168 \times 10^3}$ |
| cpusmall | $2.105 \times 10^5$ | $\mathbf{7.768 \times 10^4}$ |
| boston | $1.248 \times 10^6$ | $\mathbf{9.998 \times 10^4}$ |
| bodyfat | $1.694 \times 10^7$ | $\mathbf{2.403 \times 10^6}$ |
| eunite2001 | $8.629 \times 10^5$ | $\mathbf{2.052 \times 10^5}$ |
| pyrim | $7.667 \times 10^7$ | $\mathbf{7.523 \times 10^6}$ |

Table 2: Prediction errors.

| dataset | prediction error | |
| --- | --- | --- |
| | origin | ours |
| abalone | $\mathbf{2.232}$ | $\mathbf{2.232}$ |
| cpusmall | $\mathbf{7.886}$ | $\mathbf{7.886}$ |
| boston | $\mathbf{9.887}$ | $\mathbf{9.887}$ |
| bodyfat | $\mathbf{5.434 \times 10^{-3}}$ | $\mathbf{5.434 \times 10^{-3}}$ |
| eunite2001 | $\mathbf{2.010 \times 10^2}$ | $\mathbf{2.010 \times 10^2}$ |
| pyrim | $\mathbf{4.615 \times 10^{-3}}$ | $\mathbf{4.615 \times 10^{-3}}$ |

results are shown in Table 2. The squared losses of our approach are the same as those of the original approach. This is because our approach is guaranteed to yield the same value of the objective function as that of the original approach, as described in Theorem 2. The results presented in Table 2 indicate that the prediction results match those of the original approach while improving the efficiency.

# 6    Conclusion

We proposed a fast Block Coordinate Descent for Sparse Group Lasso. The main bottleneck of the original Block Coordinate Descent is the computation to check whether groups have zero or nonzero parameter vectors, because it uses all the parameters or data points. In contrast, our approach identifies the groups whose parameters *must be zeros* by using the parameters in the group, and skips the computation. Furthermore, the proposed approach identifies the candidate group set, which contains the groups whose parameters *must not be zeros*. The parameters are preferentially updated in the set to raise the effectiveness of Block Coordinate Descent. The attractive point of our method is that it does not need any additional hyperparameters. In addition, it provably guarantees the same results as the original method. The experimental results showed that our method reduces the processing time by up to 97% without any loss of accuracy compared with that of the original method.

## Footnotes

[1]We show all the proofs in the supplementary material.

[2]https://www.csie.ntu.edu.tw/~cjlin/libsvm/

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
