[Supplementary Material]

# Supplementary Material: Fast Sparse Group Lasso

**Yasutoshi Ida**[1,3]    **Yasuhiro Fujiwara**[2]    **Hisashi Kashima**[3,4]

[1]NTT Software Innovation Center    [2]NTT Communication Science Laboratories
[3]Kyoto University    [4]RIKEN AIP
yasutoshi.ida@ieee.org
yasuhiro.fujiwara.kh@hco.ntt.co.jp
kashima@i.kyoto-u.ac.jp

We provide the proofs of lemmas and theorems in the paper.

## A    Proof of Lemma 1

**Lemma 1 (Groups with zero vectors)** *If we have $U^{(g)} \leq \sqrt{p_g}(1-\alpha)\lambda$, parameter $\beta^{(g)}$ for group g is a zero vector.*

Before we prove the above lemma, we prove the following lemmas:

**Lemma 1.1** *For each $\|S(X^{(g)\mathrm{T}}r_{(-g)}, \alpha\lambda)\|_2$ of group g, we have $\|X^{(g)\mathrm{T}}r_{(-g)}\|_2 \geq \|S(X^{(g)\mathrm{T}}r_{(-g)}, \alpha\lambda)\|_2$.*

**Proof**    *We have*

$$\|S(z, \alpha\lambda)\|_2 = \|\mathrm{sign}(z)(|z| - \alpha\lambda)_+\|_2 = \|(|z| - \alpha\lambda)_+\|_2. \tag{A.1}$$

*In addition, we have*

$$\|(|z| - \alpha\lambda)_+\|_2 \leq \||z|\|_2 = \|z\|_2, \tag{A.2}$$

*because $0 \leq (|z| - \alpha\lambda)_+ \leq |z|$. From Equations (A.1) and (A.2), we have $\|S(z, \alpha\lambda)\|_2 \leq \|z\|_2$. Because $z = X^{(g)\mathrm{T}}r_{(-g)}$, we achieve the inequality of Lemma 1.1.* □

The following lemma shows that $U^{(g)}$ is the upper bound of $\|S(X^{(g)\mathrm{T}}r_{(-g)}, \alpha\lambda)\|_2$:

**Lemma 1.2** *For each $\|S(X^{(g)\mathrm{T}}r_{(-g)}, \alpha\lambda)\|_2$ of group g, we have $U^{(g)} \geq \|S(X^{(g)\mathrm{T}}r_{(-g)}, \alpha\lambda)\|_2$.*

**Proof**    *Let $K^{(g,:)} \in \mathcal{R}^{p_g \times p}$ be a block matrix of K that corresponds to group g. We introduce notations $R^{(g)} := X^{(g)\mathrm{T}}r_{(-g)}$ and $\tilde{R}^{(g)} := X^{(g)\mathrm{T}}\tilde{r}_{(-g)}$ for simplicity. From Equation (5), we have*

$$R^{(g)} = X^{(g)\mathrm{T}}y - K^{(g,:)}\beta + K^{(g,g)}\beta^{(g)}.$$

*This equation is transformed to the following form:*

$$X^{(g)\mathrm{T}}y - K^{(g,:)}\tilde{\beta} + K^{(g,g)}\tilde{\beta}^{(g)} + K^{(g,g)}\Delta\beta^{(g)} - K^{(g,:)}\Delta\beta$$
$$= \tilde{R}^{(g)} + K^{(g,g)}\Delta\beta^{(g)} - \sum_{l=1}^{G} K^{(g,l)}\Delta\beta^{(l)},$$

*where $\Delta\beta^{(g)} = \beta^{(g)} - \tilde{\beta}^{(g)}$ and $\Delta\beta = \beta - \tilde{\beta}$. From the aforementioned and the triangle inequality, we obtain the following inequality:*

$$\|R^{(g)}\|_2 \leq \|\tilde{R}^{(g)}\|_2 + \|K^{(g,g)}\Delta\beta^{(g)}\|_2 + \sum_{l=1}^{G}\|K^{(g,l)}\Delta\beta^{(l)}\|_2. \tag{A.3}$$

*In the second and third terms on the right-hand side of Equation (A.3), the $i$-th element of $K^{(g,l)}\Delta\beta^{(l)}$ is computed as the inner product of $K^{(g,l)}[i,:]\Delta\beta^{(l)}$. We then obtain the following upper bound for the absolute value of the inner product by using the Cauchy–Schwarz inequality:*

$$|K^{(g,l)}[i,:]\Delta\beta^{(l)}| \leq \|K^{(g,l)}[i,:]\|_2 \|\Delta\beta^{(l)}\|_2. \tag{A.4}$$

*In addition, because $\|\Delta\beta^{(l)}\|_2$ is a scalar, we obtain the following inequality for $\|K^{(g,l)}\Delta\beta^{(l)}\|_2$ in Equation (A.3) by using Equation (A.4):*

$$\|K^{(g,l)}\Delta\beta^{(l)}\|_2 \leq \|\hat{K}^{(g)}[l]\|_2 \|\Delta\beta^{(l)}\|_2 = \Lambda(g,l). \tag{A.5}$$

*From Equations (A.3) and (A.5), we obtain the following upper bound:*

$$\|R^{(g)}\|_2 \leq \|\tilde{R}^{(g)}\|_2 + \Lambda(g,g) + \sum_{l=1}^{G}\Lambda(g,l) = U^{(g)}.$$

*Finally, we obtain the following upper bound by using the aforementioned inequality and Lemma 1.1:*

$$\|S(X^{(g)\mathrm{T}}r_{(-g)},\alpha\lambda)\|_2 \leq \|X^{(g)\mathrm{T}}r_{(-g)}\|_2 = \|R^{(g)}\|_2 \leq U^{(g)},$$

*which completes the proof.* □

Now, we prove Lemma 1:

**Proof** *We have $\|S(X^{(g)\mathrm{T}}r_{(-g)},\alpha\lambda)\|_2 \leq U^{(g)} \leq \sqrt{p_g}(1-\alpha)\lambda$ from Lemma 1.2 if $U^{(g)} \leq \sqrt{p_g}(1-\alpha)\lambda$. Because $\|S(X^{(g)\mathrm{T}}r_{(-g)},\alpha\lambda)\|_2 \leq \sqrt{p_g}(1-\alpha)\lambda$ holds for group $g$, $\beta^{(g)}$ is the zero vector from Equation (4).* □

# B    Proof of Lemma 2

**Lemma 2 (Computation cost for online update scheme of upper bound)** *Given    precomputed $\|X^{(g)\mathrm{T}}\tilde{r}_{(-g)}\|_2$ and $\|\hat{K}^{(g)}[\cdot]\|_2$, we can update the upper bound of Equation (6) in $\mathcal{O}(p_g)$ time for group $g$ by using Equation (7) when the parameter vector of the group is updated.*

**Proof** *For the computation of Equation (6), we can precompute $\|X^{(g)\mathrm{T}}\tilde{r}_{(-g)}\|_2$ and $\|\hat{K}^{(g)}[\cdot]\|_2$ before entering the group-level loop and main loop, respectively. Thus, we can obtain the terms $\|X^{(g)\mathrm{T}}\tilde{r}_{(-g)}\|_2$, $\|\hat{K}^{(g)}[\cdot]\|_2$, and $\Lambda(g,l)$ in Equation (6) in $\mathcal{O}(1)$ time. If $\beta^{(g)}$ is updated to $\beta^{(g)'}$, the upper bound of Equation (6) can be updated by using Equation (7). It needs $\mathcal{O}(p_g)$ time because $\|\beta^{(g)'} - \tilde{\beta}^{(g)}\|_2$ is computed in $\mathcal{O}(p_g)$ time, and we can obtain $\Lambda(g,g)$ and $\|\hat{K}^{(g)}[g]\|_2$ in $\mathcal{O}(1)$ time, just as described. Therefore, we can compute the upper bound of Equation (6) in $\mathcal{O}(p_g)$ time for group $g$ by using Equation (7).* □

# C    Proof of Lemma 3

**Lemma 3 (Error bound)** *Let $\epsilon$ be an error bound of $C^{(g)}$ for $\|S(X^{(g)\mathrm{T}}r_{(-g)},\alpha\lambda)\|_2$ such that $|C^{(g)} - \|S(X^{(g)\mathrm{T}}r_{(-g)},\alpha\lambda)\|_2| \leq \epsilon$. We then have $\epsilon = \Lambda(g,g) + \sum_{l=1}^{G}\Lambda(g,l) + \alpha\lambda\sqrt{p_g/2}$. In addition, we have $|U^{(g)} - \|S(X^{(g)\mathrm{T}}r_{(-g)},\alpha\lambda)\|_2| \leq 2\epsilon$.*

Before we prove the above lemma, we introduce the following definitions and lemmas:

**Definition 1 (Lower bound)** *Let $L^{(g)}$ be a lower bound of $\|S(X^{(g)\mathrm{T}}r_{(-g)},\alpha\lambda)\|_2$ in Equation (4) and $\tilde{r}_{(-g)}$ be a partial residual of Equation (3) before entering the group-level outer loop. Then, $L^{(g)}$ is defined as follows:*

$$L^{(g)} = \|X^{(g)\mathrm{T}}\tilde{r}_{(-g)}\|_2 - \Lambda(g,g) - \sum_{l=1}^{G}\Lambda(g,l) - \alpha\lambda\sqrt{2p_g}, \tag{C.1}$$

*where $\Lambda(g,l) = \|\hat{K}^{(g)}[l]\|_2\|\beta^{(l)} - \tilde{\beta}^{(l)}\|_2$. In the above equation, the $i$-th element of $\hat{K}^{(g)}[l] \in \mathcal{R}^{p_g}$ is given as $\|K^{(g,l)}[i,:]\|_2$, that is, the $l_2$ norm of the $i$-th row vector in block matrix $K^{(g,l)} \in \mathcal{R}^{p_g \times p_l}$ of $K := X^{\mathrm{T}}X \in \mathcal{R}^{p \times p}$. $\tilde{\beta}^{(g)}$ is a parameter vector before entering the group-level outer loop.*

**Lemma 3.1** *For each $\|S(X^{(g)\mathrm{T}}r_{(-g)}, \alpha\lambda)\|_2$ of group $g$, we have $L^{(g)} \leq \|S(X^{(g)\mathrm{T}}r_{(-g)}, \alpha\lambda)\|_2$.*

**Proof** *Let $I_{(g)}^+ \in \{0, 1\}^{p_g}$ be a vector whose $i$-th element takes 1 if the absolute value of $i$-th element in $X^{(g)\mathrm{T}}r_{(-g)}$ is greater than $\alpha\lambda$, and takes 0 if the absolute value of the element is less than or equal to $\alpha\lambda$. We then have*

$$\|S(X^{(g)\mathrm{T}}r_{(-g)}, \alpha\lambda)\|_2 = \||X^{(g)\mathrm{T}}r_{(-g)}| - \alpha\lambda I_{(g)}^+ - |X^{(g)\mathrm{T}}r_{(-g)}| \odot (1 - I_{(g)}^+)\|_2. \qquad \text{(C.2)}$$

*According to Equation (C.2) and the triangle inequality, we have*

$$\||X^{(g)\mathrm{T}}r_{(-g)}| - \alpha\lambda I_{(g)}^+ - |X^{(g)\mathrm{T}}r_{(-g)}| \odot (1 - I_{(g)}^+)\|_2$$
$$\geq \|X^{(g)\mathrm{T}}r_{(-g)}\|_2 - \|\alpha\lambda I_{(g)}^+\|_2 - \|X^{(g)\mathrm{T}}r_{(-g)} \odot (1 - I_{(g)}^+)\|_2. \qquad \text{(C.3)}$$

*Let $p_g^+$ be the number of elements, taking the value 1 in $I_{(g)}^+$. Then, we clearly have*

$$\|\alpha\lambda I_{(g)}^+\|_2 = \alpha\lambda\sqrt{p_g^+}. \qquad \text{(C.4)}$$

*In addition, because the absolute values of the elements in $X^{(g)\mathrm{T}}r_{(-g)} \odot (1 - I_{(g)}^+)$ is less than or equal to $\alpha\lambda$, we have*

$$\|X^{(g)\mathrm{T}}r_{(-g)} \odot (1 - I_{(g)}^+)\|_2 \leq \|\alpha\lambda(1 - I_{(g)}^+)\|_2 = \alpha\lambda\sqrt{p_g - p_g^+}. \qquad \text{(C.5)}$$

*From Equations (C.2), (C.3), (C.4), and (C.5), we have*

$$\|S(X^{(g)\mathrm{T}}r_{(-g)}, \alpha\lambda)\|_2 \geq \|X^{(g)\mathrm{T}}r_{(-g)}\|_2 - \alpha\lambda\sqrt{p_g^+} - \alpha\lambda\sqrt{p_g - p_g^+}$$
$$= \|X^{(g)\mathrm{T}}r_{(-g)}\|_2 - \alpha\lambda(\sqrt{p_g^+} + \sqrt{p_g - p_g^+})$$
$$\geq \|X^{(g)\mathrm{T}}r_{(-g)}\|_2 - \alpha\lambda\sqrt{2p_g}. \qquad \text{(C.6)}$$

*We use the inequality of $\sqrt{p_g^+} + \sqrt{p_g - p_g^+} \leq 2\sqrt{p_g/2}$ in Equation (C.6). Next, we derive the lower bound of $\|X^{(g)\mathrm{T}}r_{(-g)}\|_2$ in Equation (C.6). Similar to Equation (A.3) in the proof of the upper bound, we have the following inequality by using the triangle inequality:*

$$\|X^{(g)\mathrm{T}}r_{(-g)}\|_2 = \|R^{(g)}\|_2 \geq \|\tilde{R}^{(g)}\|_2 - \|K^{(g,g)}\Delta\beta^{(g)}\|_2 - \sum_{l=1}^G \|K^{(g,l)}\Delta\beta^{(l)}\|_2. \qquad \text{(C.7)}$$

*From Equations (C.7) and (A.5), we obtain the following lower bound:*

$$\|X^{(g)\mathrm{T}}r_{(-g)}\|_2 \geq \|\tilde{R}^{(g)}\|_2 - \Lambda(g, g) - \sum_{l=1}^G \Lambda(g, l). \qquad \text{(C.8)}$$

*Finally, we obtain the following lower bound by using Equations (C.6) and (C.8):*

$$\|S(X^{(g)\mathrm{T}}r_{(-g)}, \alpha\lambda)\|_2 \geq \|X^{(g)\mathrm{T}}\tilde{r}_{(-g)}\|_2 - \Lambda(g, g) - \sum_{l=1}^G \Lambda(g, l) - \alpha\lambda\sqrt{2p_g} = L^{(g)}, \qquad \text{(C.9)}$$

*which completes the proof.* □

The following lemma shows that we can identify groups whose parameters must be nonzeros by using the lower bound:

**Lemma 3.2 (Groups with nonzero vectors)** *If we have $L^{(g)} > \sqrt{p_g}(1 - \alpha)\lambda$, parameter $\beta^{(g)}$ for group $g$ is a nonzero vector.*

**Proof** *We have $\|S(X^{(g)\mathrm{T}}r_{(-g)}, \alpha\lambda)\|_2 \geq L^{(g)} > \sqrt{p_g}(1 - \alpha)\lambda$ from Lemma 3.1 if $L^{(g)} > \sqrt{p_g}(1 - \alpha)\lambda$. Because $\|S(X^{(g)\mathrm{T}}r_{(-g)}, \alpha\lambda)\|_2 > \sqrt{p_g}(1 - \alpha)\lambda$ holds for group $g$, $\beta^{(g)}$ is the nonzero vector from Equation (4).* □

If we have $L^{(g)} \leq \sqrt{p_g}(1 - \alpha)\lambda$, it is clear that parameter $\beta^{(g)}$ for group $g$ can have a zero vector. Now, we prove Lemma 3:

**Proof** *From Lemma 1.2 and 3.1, we have*

$$L^{(g)} \leq \|S(X^{(g)\mathrm{T}} r_{(-g)}, \alpha\lambda)\|_2 \leq U^{(g)}. \tag{C.10}$$

*In addition, we also have*

$$L^{(g)} \leq C^{(g)} \leq U^{(g)}, \tag{C.11}$$

*because*

$$\frac{L^{(g)} + U^{(g)}}{2} = \|X^{(g)\mathrm{T}} \tilde{r}_{(-g)}\|_2 - \alpha\lambda\sqrt{p_g/2} = C^{(g)}. \tag{C.12}$$

*Thus, we have $L^{(g)} = C^{(g)} - \Lambda(g,g) - \sum_{l=1}^{G} \Lambda(g,l) - \alpha\lambda\sqrt{p_g/2}$, and $U^{(g)} = C^{(g)} + \Lambda(g,g) + \sum_{l=1}^{G} \Lambda(g,l) + \alpha\lambda\sqrt{p_g/2}$. Therefore, the exact value of $\|S(X^{(g)\mathrm{T}} r_{(-g)}, \alpha\lambda)\|_2$ exists within $C^{(g)} \pm \epsilon$, where $\epsilon = \Lambda(g,g) + \sum_{l=1}^{G} \Lambda(g,l) + \alpha\lambda\sqrt{p_g/2}$. In other words, we have*

$$|C^{(g)} - \|S(X^{(g)\mathrm{T}} r_{(-g)}, \alpha\lambda)\|_2| \leq \epsilon. \tag{C.13}$$

*For the upper bound, we clearly obtain the following inequality from Equation (C.10):*

$$|U^{(g)} - \|S(X^{(g)\mathrm{T}} r_{(-g)}, \alpha\lambda)\|_2| \leq |U^{(g)} - L^{(g)}| = 2\epsilon, \tag{C.14}$$

*which completes the proof.* □

## D   Proof of Lemma 4

**Lemma 4 (Groups containing nonzero vectors)** *Candidate group set $\mathbb{C}$ contains the groups whose parameters must be nonzeros.*

**Proof** *From Equation (C.11), we have*

$$L^{(g)} \leq C^{(g)}. \tag{D.1}$$

*By following Definition 4, we add a group to the candidate group set when the condition of $C^{(g)} > \sqrt{p_g}(1-\alpha)\lambda$ holds. Then, we can consider two situations for the group from Equation (D.1): (i) $C^{(g)} \geq L^{(g)} > \sqrt{p_g}(1-\alpha)\lambda$ and (ii) $C^{(g)} > \sqrt{p_g}(1-\alpha)\lambda \geq L^{(g)}$. In the case of (i), the group must have a nonzero parameter vector based on Lemma 3.2. In the case of (ii), the group can clearly have a nonzero parameter vector according to Lemma 3.2. Therefore, the candidate group set contains the groups whose parameters must be nonzeros because case (i) is included in the condition of $C^{(g)} > \sqrt{p_g}(1-\alpha)\lambda$ in Definition 4.* □

The candidate group set contains a subset of the nonzero groups *identified by using the lower bound of Lemma 3.2*. The lower bound confidently identifies groups with *nonzero* vectors while the upper bound of Lemma 1 identifies groups with *zero* vectors.

## E   Proof of Lemma 5

**Lemma 5 (Computation cost of candidate group set)** *Given precomputed $\|X^{(g)\mathrm{T}} \tilde{r}_{(-g)}\|_2$, we can extract candidate group set $\mathbb{C}$ at $\mathcal{O}(G)$ time.*

**Proof** *Equation (8) can be clearly computed at O(1) time if we pre-compute $\|X^{(g)\mathrm{T}} \tilde{r}_{(-g)}\|_2$. Therefore, we require O(G) time to extract the candidate group set because we compute Equation (8) for all the groups.* □

## F   Proof of Theorem 1

**Theorem 1 (Computation cost)** *Let $S$ and $S'$ be the rates of the un-skipped groups when Lemma 1 and Equation (4) are used, respectively. Suppose that all groups have the same size, $p_g$. If $t_m$ and $t_f$ are the numbers of iterations of BCD for the main loop and the element-level loop, respectively, our approach requires $\mathcal{O}(G\{(Q + St_m)(pp_g + p_g^2) + S'p_g t_m(t_f p_g + 1) + Q\})$ or $\mathcal{O}(G\{(Q + St_m)np_g + S'p_g t_m(t_f p_g + 1) + Q\})$ time.*

**Proof** *In Algorithm 1, first, $\|\hat{K}^{(g)}[l]\|_2$ is precomputed, which requires $\mathcal{O}(p_g)$ time. For all groups, the precomputation requires $\mathcal{O}(p)$ time. Before entering the group-level loop, $\|X^{(g)\mathrm{T}}\tilde{r}_{(-g)}\|_2$ is precomputed in $\mathcal{O}(GQ(pp_g + p_g^2))$ or $\mathcal{O}(GQnp_g)$ time to obtain the upper bounds. We can update the upper bounds when a parameter vector in a group is updated according to Definition 2. Thus, the updates of upper bounds require $\mathcal{O}(GS'p_g)$ time for a group-level loop by following Lemma 2. If the group is not skipped with respect to the upper bound, Equation (4) is computed in $\mathcal{O}(pp_g + p_g^2)$ time. Because the unskipped rate is $S$, the computation cost is $\mathcal{O}(GS(pp_g + p_g^2))$ or $\mathcal{O}(GSnp_g)$ time. If the group is not skipped with respect to Equation (4), the parameter vector is updated using Equation (2). Equation (2) requires $\mathcal{O}(p_g^2)$ time because $\|S(Z^{(g)}, t\alpha\lambda)\|_2$ in Equation (4) requires $\mathcal{O}(p_g)$ time, and $Z^{(g)}$ can be obtained in $\mathcal{O}(p_g^2)$ time based on precomputations. Because the unskipped rate is $S'$ and the number of iterations for the element-level loop is $t_f$, the computation requires $\mathcal{O}(GS't_f p_g^2)$ time. Because the number of iterations for the main loop of BCD is $t_m$, the total computation cost of our approach is $\mathcal{O}(G\{(Q + St_m)(pp_g + p_g^2) + S'p_g t_m(t_f p_g + 1) + Q\})$ or $\mathcal{O}(G\{(Q + St_m)np_g + S'p_g t_m(t_f p_g + 1) + Q\})$ time.* □

## G    Proof of Theorem 2

**Theorem 2 (Convergence property)** *Suppose that the regularization constants in Algorithm 1 are the same as those of the original BCD, and the BCD converges. Then, the solution of Algorithm 1 has the same value of the objective function as that of the original BCD.*

**Proof** *From Lemma 1, the parameter vector of group g must be a zero vector if $U^{(g)} \leq \sqrt{p_g}(1-\alpha)\lambda$ holds. Thus, we safely skip the computation of such groups. In other cases, the parameter vector can be a nonzero vector, and Algorithm 1 does not skip the computation of such groups. As the algorithm updates all the parameters until convergence after updating the parameters in the candidate group set, it will converge to the solution with the minimum value of problem (1) [2, 4, 1, 3]. Because Algorithm 1 and the standard approach are based on block coordinate descent, it is clear that the solution of Algorithm 1 has the minimum value of problem (1), the same as that of the standard approach.* □

We note that Theorem 2 holds regardless of the group updating order because it guarantees *the converged value of the objective function* rather than *the converged parameter vector*. Since the problem of SGL is either convex or strongly convex depending on the condition of *X*, the *optimal value of the objective function* is unique (while *the optimal parameter vector* may be non-unique). In addition, because we assume that the original BCD converges to the solution in the theorem, the BCD using the upper bound also converges even if it starts with the initial parameter resulting from the BCD on the candidate group set.