[Reviews · NeurIPS 2019]

Reviewer 1



In this work the author(s) focus on block coordinate descent as a method for optimizing the block-sparse version of LASSO. The main contribution is to reduce the computational cost of the iterative thresholding performed when optimizing over each block (dissected as the groups). This reduction is done via bounds on the thresholding decision boundary that are cheaper to compute, but indicate when a block is confidently not zero, or confidently all zero. I have a number of questions about the method as outlined below: 1) This method seems to only be applicable to disjoint groups of coefficients. Many applications require overlapping groups (e.g., Group sparse optimization by alternating direction method by Deng, Yin, and Zhang). Does this method trivially extend? It does not seem to, at least to me. 2) This method focuses only on increasing the efficiency of the block-coordinate descent. Other methods, such as iterative thresholding for group-sparse inference, which does not use a full internal block iteration and instead only performs one step of thresholding per block, can be used instead (e.g., Iterative Thresholding for Sparse Approximations by Blumensath and Davies and others since then). Do any of the benefits transfer? Depending on the number of iterations it takes to compute, these methods could be comparable, or be much slower. It would be useful for the authors to compare to these other methods Minor comments: Line 60: Solution \hat{\beta} is obtained --> The solution \hat{\beta} is obtained Line 122: this norm is broken up over 2 lines: ||K^(g,l)[i,:]||_2

Reviewer 2



Originality: this paper proposes a new bound on the screening criterion that help to reduce the computation of the group Lasso. Combined with block coordinate descent, experiments show a drastic improvement compare to recent screening methods. Quality: the mathematical analysis relies on previous results on screening methods. Though no sketches are given for the most important lemmas and theorem (they all are in the supplementary material), the guiding line is clear and motivated. Clarity: the paper is easy to read. All the idea are motivated and the algorithm come easily. Significance: as group Lasso is a very used and useful methods when dealing with high dimensional data and small dataset, such work is significant as it propose a pretty fast method.

Reviewer 3



Summary: This paper presents a fast block coordinate descent algorithm for the sparse-group lasso problem. Two strategies are proposed to improve the computational efficiency. The first strategy is quickly identifying the groups of inactive features. The idea is to use an easy-to-compute upper bound when checking if the inactive-group condition holds. The second strategy is to select a set of candidate groups and update the feature vectors inside those groups first before iterating over all groups. Empirical comparisons are conducted to show that the proposed method has faster running time than other methods. Strength: Overall the paper is well-written and easy to follow. The proposed algorithm seems new and promising. Weakness: The proof of Theorem 2 is not very clear to me. Theorem 2 says that the proposed algorithm converges to a global minimum if using the same sequence of regularization constants as the original BCD algorithm. Its proof (shown in the appendix) simply uses the fact that the proposed algorithm safely skips the groups that must be zeros, and that each group is updated at least once in every loop. This proof seems to ignore a major difference in the group updating order between the proposed algorithm and the original BCD algorithm: the proposed algorithm first updates the candidate groups and then iterates over all groups, while the original BCD algorithm directly iterates over all groups. Note that the candidate group set does not contain ALL groups that must be non-zeros (it only contains a subset of the those groups, see my detailed comments below). Because the difference in the group updating order may cause a difference in the convergence behavior, the proof of Theorem 2 needs to justify that the proposed algorithm can still converge if the same sequence of regularization constants is used as in the original BCD algorithm. Besides the convergence property in Theorem 2, this paper does not give a convergence rate of the proposed algorithm, and compare that with the original BCD algorithm. One related question: the statement of Lemma 4 is confusing to me. Does Lemma 4 mean that the candidate group set contains *ALL* groups whose parameters must be nonzeros? However, this is not true because it can happen that a nonzero group (i.e., condition (4) does not hold) is *NOT* in the candidate set (i.e., condition (9) does not hold). Therefore, the candidate group set only contains a subset of the nonzero groups. -------------After reading the rebuttal-------------- The authors have addressed my concerns on the proof of Theorem 2 and the statement of Lemma 4. The analysis of the convergence rate is probably hard because of the bias in the update rule. The techniques are practical and I am happy to increase my score.

[Author Response · NeurIPS 2019]

First of all, we would thank all the reviewers for their careful reading and thoughtful comments. We address their comments and questions below. We will prepare our final version based on their comments.

——————- Response to Reviewer 1: ——————-

**- This method seems to only be applicable to disjoint groups of coefficients. Many applications require overlapping groups (e.g.,...). Does this method trivially extend? It does not seem to, at least to me.**

Answer: Our method is relatively easy to be extended for overlapping groups by using overlap norm, which associates to the parameter vector a specific decomposition [a]. Since we can still use BCD for the overlap norm with a slight modification, our method is also available to the BCD with the overlapping group.

**- This method focuses only on increasing the efficiency of the block-coordinate descent. Other methods, such as iterative thresholding for group-sparse inference, which does not use a full internal block iteration and instead only performs one step of thresholding per block, can be used instead (e.g.,...). Do any of the benefits transfer? Depending on the number of iterations it takes to compute, these methods could be comparable, or be much slower. It would be useful for the authors to compare to these other methods**

Answer: Yes, our ideas of the upper bound and the candidate group set is applicable to other methods such as the iterative thresholding with a slight modification of Eq. (6) and Eq. (8). This is because the computations of the bound and the set are based on the difference between the reference parameter vector and the current parameter vector, which can also be handled in the iterative thresholding.

**- I would like to see comparisons to other**

We compared our method with an iterative thresholding-style algorithm. We shows the result for $\alpha = 0.2$ in Figure 1 as a example. Our method outperformed Iterative Thresholding. The similar results have been observed for other $\alpha$. We will add all the results to the final version of our paper. We thank the reviewer for this constructive comment.

——————- Response to Reviewer 2: ——————-

**- a figure giving the ratio of activated variable could be useful to compare the selection process (see fig. 4 in [1])**

Figure 2 shows the result of the selection process on the boston dataset by following Figure 4 in [1]. $K$ represents the number of the main loop of BCD using the upper bound. Our method effectively skips the variables even if $\lambda$ is small. We will add the results of other datasets in the final version of our paper. We appreciate your insightful advice.

**- clarify at the beginning that the method only consider non-overlapping groups**

We will revise the paper to reflect this comment. We thank the reviewer for your important advice.

**- what are some possible extension of the proposed algorithm ?**

Our method can be flexibly extended to various types of group structured data. For examples, our method can be naturally extended to overlapping groups by using overlap norm [a]. In addition, our method is also applicable for tree-structured data since Tree-structured Group Lasso [b] uses a similar equation to Eq. (4) of SGL.

——————- Response to Reviewer 3: ——————-

**- Provide a more clear proof of Theorem 2. (Related comment: This proof seems to ignore a major difference in the group updating order...)**

Theorem 2 holds regardless of the group updating order because it guarantees *the converged value of the objective function* rather than *the converged parameter vector*. Since the problem of SGL is either convex or strongly convex depending on the condition of $X$, the *optimal value of the objective function* is unique (while *the optimal parameter vector* may be non-unique). In addition, because we assume that the original BCD converges to the solution in the theorem, the BCD using the upper bound also converges even if it starts with the initial parameter resulting from the BCD on the candidate group set. We will add this explanation to the paper in detail.

**- Derive a convergence rate and compare that with the original BCD algorithm.**

Our method performs the first BCD on the candidate group set and then performs the second BCD on all the groups using the upper bound. We would like to analyze the first BCD in the future work because our own criterion of Eq. (8) is nontrivial for the convergence analysis. However, since the updating order of the second BCD is the same as the original BCD, the second BCD achieves at least the same convergence rate as that of the original BCD with the initial parameter vector resulting from the first BCD.

**- State Lemma 4 in a more clear way. (Related question: Does Lemma 4 mean that the candidate group set contains *ALL* groups whose parameters must be nonzeros?)**

No, the candidate group set contains a subset of the nonzero groups *identified by using the lower bound of Lemma 3.2* as shown in the proof of Lemma 4. The lower bound confidently identifies groups with *nonzero* vectors while the upper bound of Lemma 1 identifies groups with *zero* vectors. We are sorry for the confusion. We will modify Lemma 4 to add the aforementioned explanation.

[a] L. Jacob, G. Obozinski, J. Vert. Group lasso with overlap and graph lasso. In ICML, 2009.

[b] R. Jenatton, J. Mairal, G. Obozinski, F. Bach. Proximal Methods for Sparse Hierarchical Dictionary Learning. In ICML, 2010.

Figure 1: Comparison with iterative thresholding.($\alpha = 0.2$)

Figure 2: The ratio of active variables during the optimization of our method. $K$ represents the number of the main loop of BCD using the upper bound. ($\alpha = 0.2$)

[Meta-Review · NeurIPS 2019]

The authors study a new block coordinate descent algorithm for the sparse group lasso. The contributions appear solid and the reviewers were mostly positive about the work. I would like the authors to pay close attention to the points raised by Reviewer 3 about the theory and to revise the paper so that this is made more clear to the reader. Also, if the proposed algorithm applies to the overlap case (a very important case in general) then this should also be made clear.